# Effects of Different Feed Additives on Intestinal Metabolite Composition of Weaned Piglets

**DOI:** 10.3390/metabo14030138

**Published:** 2024-02-26

**Authors:** Mingxuan Zhao, Jian Zhang, Fuzhou Liu, Lv Luo, Mingbang Wei, Yourong Ye, Chamba Yangzom, Peng Shang

**Affiliations:** 1College of Animal Science, Tibet Agriculture and Animal Husbandry University, Linzhi 860000, China; m19863998873@163.com (M.Z.); zhj88n@126.com (J.Z.); liufuzhou2022@163.com (F.L.); ll202300201095@163.com (L.L.); bangbangbang962@163.com (M.W.); yeyourong@xza.edu.cn (Y.Y.); 2The Key Laboratory of Tibetan Pigs Genetic Improvement and Reproduction Engineering, Linzhi 860000, China; 3The Provincial and Ministerial Co-Founded Collaborative Innovation Center for R & D in Tibet Characteristic Agricultural and Animal Husbandry Resources, Linzhi 860000, China

**Keywords:** weaned piglets, metabolomics, Tibetan medicine, diarrhea, growth performance

## Abstract

To study the effects of different feed additives on the weaning stress of Tibetan piglets, we selected 28 healthy, 30-day-old Tibetan weaned piglets and divided them into four groups, namely, the control group (basal feed without any antibiotic additions) (Nor), the group with the addition of the antibiotic lincomycin (Ant), the group with the addition of fifteen-flavor black pills of Tibetan medicine (Tib), and the group with the addition of fecal bacterial supernatant (Fec). We measured growth performance, blood physiological indexes, and metabolomics. The results showed that the Ant, Tib, and Fec groups significantly reduced the ratio of diarrhea to feed/weight (F/G) and increased the average daily gain (ADG) compared with the Nor group (*p* < 0.01). The Nor group had significantly lower leukocyte counts, hemoglobin levels, and erythrocyte counts compared with the other three groups at 21 d (*p* < 0.05). These physiological indexes tended to stabilize at 42 d. We found that there were beneficial metabolites and metabolic pathways for gastrointestinal function. Specifically, the porphyrin metabolic pathway was elevated in the Ant group, and the tryptophan metabolic pathway was significantly elevated in the Tib and Fec groups compared with the Nor group (*p* < 0.05). In conclusion, adding fecal bacterial supernatant and fifteen-flavor black pills of Tibetan medicine to the feed reduced the rate of diarrhea and improved the growth performance of the piglets. Moreover, it had an effect on the microorganisms and their metabolites and pathways in the gastrointestinal tract of the animals, which might be the main reason for influencing the diarrhea rate of weaned Tibetan piglets and the growth and development of the piglets. This study provides a new approach for anti-stress applications in weaned Tibetan piglets and the development of substitute anti-products.

## 1. Introduction

One of the most stressful events in the lifecycle of pigs is the weaning of piglets from the sow [1]. During the weaning process, piglets are exposed to multiple stressors, such as transportation, handling, transitioning from highly palatable milk to a less palatable dry feed, adjusting to a new environment, and establishing social hierarchy [2]. These stressors, along with the physiological and anatomical adaptations in the digestive tract, can lead to digestive dysfunctions and impaired immune systems, ultimately resulting in reduced growth and health in weaned nursery pigs [2]. In addition to optimizing the piglet transfer process; implementing scientific weaning techniques; reducing stress; and paying attention to temperature, humidity control, and strict disinfection of the barn after weaning [3], the addition of antibiotics to the feed is a common method to prevent and control diarrhea in piglets. However, this practice can lead to the development of antibiotic-resistant bacteria, which can be transferred from animals to humans, posing a threat to human health [4]. Consequently, the development of efficient antibiotic replacement products has become the focus of research.

Fecal microbiota transplantation (FMT) is a technique in which the fecal microbiota from a healthy individual are transplanted into the gut of a recipient individual [5,6]. FMT has garnered attention as an important therapy for intestinal diseases, including inflammatory bowel disease and irritable bowel syndrome [7,8,9]. This technique has the ability to restore the intestinal microbiota, making it a promising approach for researchers [10,11,12]. Additionally, natural herbal medicines have also gained attention due to their low side effects, multi-targeting, and multi-pathway advantages [13]. The Tibetan medicine fifteen-flavor black pill (Tibetan medicine name: Tamen Jiuaribu) prescription was developed in the 14th century BC and belongs to Chinese patent medicine preparation. The medicine is prepared from mirabilite, salt (parched), Rhododendron fortunei, nutmeg, Akebia stem, coriander fruit, mirabilite, sal ammoniac,light, and fifteen kinds of pure natural mineral and plant medicines such as salt, purple sal, bangga, Tibetan costustoot, long pepper, black pepper, and dried ginger. These pills are commonly used to treat chronic gastroenteritis, indigestion, loss of appetite, vomiting, and diarrhea [13]. The metabolites produced by different feed additives significantly impact the structure of the intestinal flora and play a crucial role in stress-induced diarrhea in weaned piglets [14]. The interaction between the metabolism of the intestinal flora and the nutrient metabolism of the host contributes to the maintenance of the intestinal tract and the overall health of the host [14]. Metabolites provide detailed information about the biological endpoints as downstream components of gene regulatory networks and protein interaction networks [15].

Currently, several studies have demonstrated the regulatory effects of probiotics, microecological agents, and antimicrobial peptides on intestinal flora. These effects include inhibiting the growth of harmful microorganisms, promoting the feed intake and growth of pigs, and improving immune function [16,17]. Such characteristics make them promising alternatives to antibiotics. Fecal transplantation has the ability to reshape the composition of the intestinal flora. In this regard, studies have shown that transplanting fecal microbiota from healthy donors to recipients with intestinal flora disorders can stabilize the intestinal microbiota and improve intestinal barrier function [18,19,20]. Research has also shown that different pig breeds can transfer their intestinal characteristics to germ-free mice through the transplantation of pig fecal fluids [21]. Moreover, recent findings indicate that the fifteen-flavor black pill of Tibetan medicine has the potential to protect the gastric mucosa, improve gastric muscle movement, promote gastric emptying, and accelerate small-intestine propulsion. Apart from these mechanisms, an increase in blood flow to the gastric mucosa of gastritis patients has been observed, along with improved microcirculation and enhanced absorption of proliferative lesions. However, the impact of the fifteen-flavor black pill of Tibetan medicine on the intestinal flora remains unexplored. Fecal metabolites, which are the end products of cellular and intestinal flora metabolism, offer a comprehensive reflection of the intestinal flora as well as the absorption and digestion of nutrients in the digestive tract. They not only elaborate on the composition of the intestinal microbiota, but also serve as biomarkers to study the relationship between intestinal microbial metabolism and host phenotype [22]. Post-weaning presents a major challenge for pigs as they are exposed to pathogenic bacteria and experience nutritional, environmental, and immunological stresses [23]. Enhancing the health condition of piglets is not only cost-effective but also advantageous for sustainable pork production. Therefore, we propose the utilization of metabolomics technology to analyze the effects of feeding four experimental groups of weaned Tibetan piglets without antibiotics and with the addition of fecal bacterial supernatant, the fifteen-flavor black pill of Tibetan medicine, and lincomycin, respectively. 

This analysis aims to investigate the impact of feeding fecal bacterial supernatant and the fifteen-flavor black pill of Tibetan medicine on the growth performance, blood physiological indexes, and metabolites of weaned Tibetan piglets. Ultimately, this study aims to contribute to the comprehensive understanding and practical implementation of feed additives in livestock production. Consequently, it intends to provide a theoretical basis for the utilization of feed additives in animal husbandry.

## 2. Materials and Methods

### 2.1. Animals, Design, and Diets

Twenty-eight weaned Tibetan pigs, aged 30 days with an initial body weight (BW) of 5.4 ± 0.2 kg, were allocated into four dietary treatments using a randomized complete block design. The trial consisted of four treatment groups: the control group (Nor1/2 group), the lincomycin group (Ant1/2 group), the fecal bacterial supernatant group (Fec1/2 group), and the Tibetan medicine fifteen-flavor black pill group (Tib1/2 group). Group 1 represents the first stage of feeding (0–21 d), and group 2 represents the second stage of feeding (21–42 d).

The Tibetan medicine 15-flavor black pill belongs to Chinese patent medicine preparation and is composed of 75 g of cold stone, 75 g of salt, 75 g rhododendron, 15 g nutmeg, 75 g of clematis, 50 g coriander fruit, 20 g fire nitrate, 20 g bright salt, 50 g herba aconiti tangutici, 50 g Tibetan combination, 25 g piper, 25 g black pepper, 40 g dried ginger, and 50 g honey, after drying. In the control group (Nor), piglets were fed a basal diet, while the Fec group, the Tib group, and the Ant group were fed the same basal diet but supplemented with 2000 mL/kg of fecal bacterial supernatant, 1000 mg/kg of Tibetan medicine fifteen-flavor black pill, and 1000 mg/kg of lincomycin hydrochloride, respectively (Table 1).

For the nutrient compositions and experimental diets, refer to Table 1. All experimental diets were formulated to meet or exceed the nutrient requirements recommended by the NRC [24]. The pigs were fed these diets for a duration of 42 d, which were divided into two phases: phase 1 (0 d to 21 d) and phase 2 (21 d to 42 d). The end day of phase 1 is also the start day of phase 2. 

### 2.2. Growth Performance

Individual body weight (BW) was recorded at the beginning and end of each phase (0 d and 42 d) to track average daily gain (ADG), average daily feed intake (ADFI), and feed-to-weight ratio (F/G) throughout the experiment. The diarrhea of the weaned piglets was checked every day.

### 2.3. Sample Collection and Processing

Fresh fecal samples were collected using rectal massage techniques from the pigs on 0 d, 21 d, and 42 d of the experiment. (Fasting for 8 h before blood collection cannot stop water and keep the original feeding environment.) The collected fecal samples were freeze-dried and ground (60 Hz, 30 s), and 50 mg of each sample was weighed into a centrifuge tube. Then, 700 μL of the extraction solution (methanol–water, 3:1 by volume, pre-cooled at −40 °C, containing the internal standard) was added. The mixture was vortexed for 30 s, homogenized for 4 min at 35 Hz, sonicated in an ice-water bath for 5 min, homogenized, and sonicated three more times. After that, the samples were kept in the homogenizer at 4 °C overnight. Subsequently, the samples were centrifuged at 4 °C, 12,000 rpm (centrifugal force 13,800× *g*, radius 8.6 cm) for 15 min, and the supernatant was carefully filtered through a 0.22 μm microporous membrane. The supernatant was then diluted 5-fold with the extraction solution and vortexed for 30 s. A QC sample was prepared by taking 40 μL of each sample and stored at −80 °C until analysis.

Blood samples were collected from the anterior vena cava of the piglets while in a fasting state on 0 d, 21 d, and 42 d of the study period. The blood samples were added to an EDTA vacuum tube (purple top) for subsequent analysis. Protein, glucose, white blood cell (WBC), red blood cell (RBC), hemoglobin (HGB), and lymphocyte concentrations in the whole blood samples were analyzed using an automatic blood analyzer (BC-2600Vet Veterinary Fully-Automated Blood Cell Analyzer, Wuxi Xuanye Science and Technology Development Co., Wuxi, China).

### 2.4. LC-MS Analysis

In this experiment, the target compounds were separated through chromatography using a Waters UPLC liquid chromatography column and an EXIONLC system (SCIEX) ultra-high performance liquid chromatograph (UPLC). The mass spectrometry data acquisition and quantitative analysis of target compounds were performed using SCIEX Analyst WorkStation software (Version 1.6.3). The raw mass spectra were converted to TXT format using MSconverter software (Version 16.2.0.9). Peak extraction and annotation were conducted using a self-developed R package and a self-constructed database. This was followed by performing orthogonal partial least squares discriminant analysis (OPLS-DA). The model validity was evaluated using the R2Y and Q2 parameters. Differential metabolites were identified based on the variable importance in the projection (VIP) of the OPLS-DA analysis between the two groups. A VIP value greater than 1 was used as a screening criterion, combined with the *p* value of the univariate analysis. Two ionization modes, positive ion mode (POS) and negative ion mode (NEG), were employed for metabolite detection in this experiment to improve metabolite coverage and detection effectiveness. The subsequent metabolite data analysis includes OPLS-DA of both the positive and negative ion mode data.

### 2.5. Mass Spectrometer Parameters

LC-MS/MS analysis was conducted using the SCIEX 6500 QTRAP+ triple quadrupole mass spectrometer, which was equipped with an IonDriveTurbo VESI ion source. The experiment was carried out in the multiple reaction monitoring (MRM) mode. The ion source parameters were set as follows: ion spray voltage, +5500/−4500 V; curtain gas pressure, 35 psi; temperature, 400 °C; ion source gas 1 pressure, 60 psi; ion source gas 2 pressure, 60 psi; and DP, ±100 V. The identical ion source parameters were utilized for the study.

### 2.6. Preprocessing and Quality Control of Raw Sequencing Data

The constructed sequencing libraries were sequenced on the Illumina NovaSeq high-throughput sequencing platform using the whole-genome birdshot method. In order to assess the stability of the analytical system and identify variables that exhibited considerable variation during the analysis process, all samples were analyzed as quality control (QC) samples. A QC sample was included every 8–10 samples during instrumental analysis. Because metabolome data are multidimensional and some variables are highly correlated, traditional univariate analysis cannot quickly, fully, and accurately mine the potential information in the data. Therefore, the OPLS-DA multivariate statistical method was applied to analyze the metabolome data, and the collected multidimensional data were reduced and classified.

### 2.7. Statistical Analysis

We performed Duncan’s multiple comparisons using SPSS 18.0 software. Growth data and blood physiological indexes of weaned Tibetan piglets were analyzed using one-way ANOVA to determine differences between treatment groups. Differences with *p* < 0.05 were considered significant, differences with *p* < 0.01 were considered highly significant, and differences with *p* > 0.05 were considered not significant.

## 3. Results

### 3.1. Growth Performance

The effects on growth performance parameters from 0 d to 42 d are presented in Table 2, Table 3 and Table 4. The incidence of diarrhea was significantly lower (*p* < 0.01) in the Ant, Tib, and Fec groups compared with the Nor group. When comparing the ADG among the four groups, the Ant, Tib, and Fec groups had significantly higher ADG values (*p* < 0.01) than the Nor group. Additionally, the Ant and Tib groups showed significantly higher ADG values (*p* < 0.05) than the Fec group. However, there was no significant difference (*p* > 0.05) observed in the comparison of ADFI among the four groups. Comparing the feed conversion ratio (F/G) among the four groups, the Nor group had the highest value, and it was significantly higher (*p* < 0.01) than the three other groups (Ant, Tib, and Fec). Furthermore, the Fec group had significantly higher F/G values (*p* < 0.01) than the Ant and Tib groups, with no significant difference between the Ant and Tib groups (*p* > 0.05).

### 3.2. Comparison of Blood Physiological Indicators

In the comparison of the blood physiological indices examined in this experiment under the same rearing conditions and at the same time of the experiment, on day 21 (Table 5), the leukocyte count in the Tib1, Fec1, and Ant1 groups was significantly higher than that in the Nor1 group (*p* < 0.01). Hemoglobin (HGB) levels were the lowest in the Nor1 group and significantly higher in the Ant1, Tib1, and Fec1 groups compared with the Nor1 group (*p* < 0.05). The analysis of blood physiological indices in weaned piglets on day 42 of the experiment yielded no significant differences in any of the blood physiological indices (*p* > 0.05), including platelet count, which did not reach statistical significance (Table 6).

### 3.3. Quality Control

When conducting metabolomics research using mass spectrometry, quality control (QC) samples are typically utilized to ensure the reliability and high quality of the obtained metabolomics data. Although the QC samples are theoretically identical, there are usually systematic errors introduced during the processes of sample extraction, detection, and analysis. These errors lead to discrepancies between the QC samples, with smaller differences indicating higher method stability and better data quality. The visualization of the PCA analysis graph for this experiment reveals a dense distribution of the QC samples, indicating the reliability of the data (Figure 1A,B). The starting group of the pre-test is referred to as the STA group.

### 3.4. Multivariate Statistical Analysis

R^2^X, R^2^Y, and Q^2^ are commonly used metrics for this purpose. In general, R^2^Y and Q^2^ values greater than 0.5 indicate the reliability of the model. As presented in Table 7, the OPLS-DA model established in this experiment exhibits R^2^Y and Q^2^ values exceeding 0.5, thus signifying the stability, reliability, and statistical significance of the model.

### 3.5. Screening for Differential Metabolites

Differential metabolites were identified based on three parameters: VIP, FC, and *p*-value. To be considered differential metabolites, the VIP value had to be greater than 1 and the *p*-value had to be less than 0.01. Furthermore, specific differential metabolites were selected based on the principle of *p*-value less than 0.01, VIP value greater than 1, and FC value greater than 1 or less than 1. These criteria were determined using the VIP values obtained from statistical analyses and OPLS-DA.

At the 21st day of the experiment for the comparisons “Nor1 vs. Ant1”, “Nor1 vs. Tib1”, and “Nor1 vs. Fec1”, a total of 41, 46, and 22 differential metabolites were identified, respectively, with 15, 19, and 9 being up-regulated and 26, 27, and 13 being down-regulated (Table 8, Table 9 and Table 10). At the 42nd day of the trial, in comparison with the control groups “Nor2 vs. Ant2”, “Nor2 vs. Tib2”, and “Nor2 vs. Fec2”, there were, respectively, 9, 5, and 17 up-regulated metabolites and 5 and 12 down-regulated metabolites (Table 11, Table 12 and Table 13). Specifically, 2, 5, and 12 metabolites were up-regulated, while 7, 0, and 5 were down-regulated, respectively.

All samples and relevant data underwent distance matrix calculation, and hierarchical clustering was performed to generate a dendrogram illustrating the similarity between the samples. The sample dendrograms revealed significant metabolic differences between different feed additives in both positive and negative ion modes (Figure 1C,D).

### 3.6. KEGG Analysis of Differential Metabolites

Pathway analysis bubble plots were generated to illustrate the metabolic pathways involved in different feed additions for the Nor and experimental groups. At the 21st day, the Ant1 group screened a total of eight metabolic pathways, including bile secretion, biosynthesis of alkaloids derived from the shikimate pathway, taste transduction, synaptic vesicle cycle, secondary bile acid biosynthesis, and gap junctions (Figure 1F). Comparatively, the Tib1 group screened a total of 11 metabolic pathways compared with the Nor1 group (Figure 1G), which included purine metabolism, tryptophan metabolism, chromogranin metabolism, and ABC transporters. The Fec1 group screened 10 metabolic pathways (Figure 1E), including tryptophan metabolism, melanogenesis, the prolactin signaling pathway, the tyrosine signaling pathway, and the gap junction pathway. At the end of the experiment, the Ant2 group exhibited 10 differential metabolic pathways compared with the Nor2 group (Figure 1I), including porphyrin metabolism, aminoacyl-tRNA biosynthesis, and amino acid biosynthesis. The Tib2 group showed five differential pathways (Figure 1J), including tryptophan metabolism, melanogenesis, the prolactin signaling pathway, and tyrosine metabolism. Furthermore, Figure 1E displays five differential pathways, with the major ones being phenylalanine metabolism, glyoxylate and dicarboxylate metabolism, and lysine degradation. The Fec2 group exhibited three metabolic pathways (Figure 1H), with the major ones being tyrosine metabolism, microbial metabolism in diverse environments, and phenylalanine metabolism.

### 3.7. Metabolite KEGG Enrichment Circle Plot Analysis

The top 20 enriched pathways were selected to draw KEGG enrichment circle maps, and it can be seen from Figure 2A that the benzoic acid family (ko07110) and serotonin receptor agonist (ko07211) were the most significant difference maps in the Nor1 and Ant1 groups. As can be seen from Figure 2B, fatty acid degradation (ko00071) was the most significant difference plot in the Nor2 and Ant2 groups. As can be seen from Figure 2C, oocyte meiosis (ko04114) and progesterone-mediated oocyte maturation (ko04914) were the most significant difference maps in the Nor1 and Fec1 groups. Figure 2D shows that fatty acid degradation (ko00071) and α-linolenic acid metabolism (ko00592) were the most significant difference maps in the Nor2 and Fec2 groups. The fatty acid degradation pathway can generate ATP as well as acetyl coenzyme A, which is one of the main sources of energy for the body, and improve the metabolism of the weaned piglets, which in turn promotes growth and development. From Figure 2E, it can be seen that the serotonin receptor agonist/antagonist (ko07211) is the most significant difference between the Nor1 and Tib1 groups. It has been shown that 95% of serotonin is located in the intestinal tract of animals, and the serotonin receptor agonist can regulate the release of serotonin, which is importantly related to maintain a good sleep and appetite. From Figure 2F, it can be seen that in the Nor2 and Tib2 groups, naphthalene (ko07114) and the phosphatidylinositol signaling system (ko04070) are the most significant differences between the graphs.

## 4. Discussion

Based on metabolomics, this experiment utilized multivariate statistical analysis to observe and analyze the effects of adding lincomycin, the fifteen-flavor black pill of Tibetan medicine, and fecal bacterial supernatant to the feed on the growth performance, blood physiological indexes, and metabolites of weaned Tibetan piglets using LC-MC technology. The results indicated that the fifteen-flavor black pill group and the fecal bacterial supernatant group had more significant effects on the fecal metabolism of the weaned piglets. The differential metabolite enrichment analysis revealed that the Tibetan medicine group was involved in the widest range of metabolic pathways on day 21 of the experiment.

Fecal microbial transplantation (FMT) is widely used to alter and remodel the gut microbiota of animals, promoting growth and development [24,25]. Oral administration of fecal microbial suspensions to adult gilts has been shown to significantly increase their average daily weight gain and reduce the rate of diarrhea in their piglets [26]. Ren et al. [27] conducted an experimental study on the growth and intestinal health of neonatal piglets, showing that FMT led to greater weight gain and body fat deposition in piglets through the introduction of fecal microorganisms into the colon. Hu et al. [28] demonstrated that transplantation of fecal microbial fluids from healthy adult Jinhua pigs to newborn piglets of “Duchangda” piglets resulted in a significant increase in the average daily gain (ADG) of the piglets. These findings closely align with the results of the present experiment, where the feeding of fecal bacterial supernatant significantly reduced the diarrhea rate of weaned piglets and moderately improved their growth performance. Blood, as a crucial component of the animal circulatory system, can objectively reflect the physiology and metabolism of the organism. Changes in the number of leukocytes can indicate the level of cellular immunity [29,30]. The Tib group of weaned piglets exhibited significantly higher leukocyte counts than the Nor group on the 21st day of the test (*p* < 0.01). Furthermore, higher hemoglobin levels contribute to enhanced oxygen transport, immune ability, and metabolic function [31,32,33]. Clinically, higher hemoglobin levels have been associated with improved body resistance [34]. In this experiment, the Tib group had significantly higher hemoglobin (HGB) levels than the Nor group (*p* < 0.05), which indicates an enhancement in the immune ability of the weaned piglets.

In addition to the fecal bacteria refeeding method, the use of natural Chinese herbs to prevent and control piglet diarrhea has gained increasing attention [35,36,37]. Traditional Chinese herbal medicines, including Tibetan medicines, have been found to be effective in treating bacterial infectious diseases [38]. In this study, the addition of the Tibetan medicine fifteen-flavor black herbal pills to the diet reduced the frequency of diarrhea in the weaned Tibetan piglets compared with the Nor group. Previous studies have also reported positive effects of dietary supplementation with herbal extracts on the frequency of diarrhea in weaned piglets [39,40]. For instance, Xia et al. [41] used the traditional Tibetan medicine Anzhi Jinhua San in mice and found it effective in treating dyspepsia, anorexia, cold, and diarrhea. Additionally, clinical studies have demonstrated the effectiveness of the Tibetan medicine fifteen-flavor black herbal pill in treating pain-oriented intermittent diarrhea patients, and in this experiment, it was added to the basal diets of weaned piglets for the first time. The results showed a significant reduction in the diarrhea rate of weaned piglets compared with the Nor group during the 21 d and 42 d test periods, with no significant difference in growth performance between the Tibetan medicine group and the lincomycin group. Moreover, the Tib group exhibited extremely significantly higher white blood cell counts than the Nor group (*p* < 0.01) and significantly higher hemoglobin (HGB) counts than the Nor group (*p* < 0.05) on day 21 of the test. These findings suggest that the fifteen-flavor black pill of Tibetan medicine has the potential to reduce the diarrhea rate of Tibetan piglets, improve their immune ability, and promote their growth.

Weaning stress disrupts the intestinal flora, leading to a decrease in the growth performance and immunity of the pig [42,43]. The addition of lincomycin, Tibetan medicine’s fifteen-flavor black pill, and fecal bacterial supernatant can enhance the growth performance and immunity of weaned piglets, although the effect on the intestinal flora still requires further investigation. Therefore, this experiment investigated the effects of different feed additives on the microbial metabolites of the intestinal flora. It was found that the Fec group, compared with the Nor group, exhibited up-regulated differential metabolites including lactitol, L-arginine succinate, and 3-hydroxy-o-cyanoanthranilic acid (3-HAA). 3-HAA, a tryptophan metabolite produced through the indoleamine-2,3-dioxygenase (IDO) pathway, plays a vital role in the regulation of the immune system [44]. Moreover, 3-HAA inhibits cytokine production by Th1 and Th2 cells through the enhancement of heme oxygenase-1 (HO-1) [45] and the expression of inducible nitric oxide synthase (iNOS) [46]. Additionally, the regulatory mechanism of 3-HAA has been shown to inhibit the production of inflammatory mediators by macrophages through the LPS-induced activation of the PI3K/Akt signaling pathway [47]. This experiment found that the Fec group modulated the immune system of weaned piglets by up-regulating the differential metabolite 3-HAA, resulting in an increase in the hemoglobin content of the piglets’ blood, inhibition of cytokine secretion by T cells, and alteration of macrophage activity. It also exhibited anti-inflammatory activity, leading to a decrease in the diarrhea rate compared with the Nor group.

According to the differential metabolite KEGG analysis, the Tib, Fec, and Nor groups were found to exhibit tryptophan metabolism, which is the main metabolic pathway in the gastrointestinal tract of animals [48]. Dietary tryptophan is primarily acted upon by gut microbes, producing indole and its derivatives. Indole and its derivatives play a protective role for bacteria against antibiotics by regulating flagellar synthesis and the expression of virulence factors [49,50,51]. They also act as agonists of the aryl hydrocarbon receptor (AhR), participating in the regulation of gut microbial load, structure, intestinal endothelium, lymphocyte immune tolerance, and other physiological processes [52]. Bansal et al. [53] demonstrated that indole promotes the expression of genes related to the mucosal barrier and mucus secretion. Moreover, a reduced expression of AhR receptors in the intestinal mucosa of Crohn’s disease patients was found, worsening symptoms of colitis due to a lack of tryptophan in the diet. Jennis et al. [54] observed increased intestinal mucosal permeability in mice that were fed a high-fat diet for 22 consecutive weeks. However, this increase was restored after treatment with indole-3-acetic acid, along with a decrease in circulating lipopolysaccharide (LPS) in the blood. In this experiment, the addition of Tibetan medicine’s fifteen-flavor black pill and fecal bacterial supernatant regulated the intestinal microflora through the tryptophan metabolic pathway, thereby increasing the bacteriostatic activity of the intestine compared with the Nor group. The AhR signaling generated by the metabolism of tryptophan along the pathway of the intestinal flora was essential for the intestinal stability of the weaned piglets [54,55]. It helped to maintain the intestinal mucosal barrier function and reduce the diarrhea rate of the weaned piglets.

In conclusion, this experiment systematically observed the effects of Tibetan medicine’s fifteen-flavor black pill and the supernatant of fecal bacteria on the intestinal metabolites of weaned Tibetan piglets for the first time. The findings confirmed that refeeding with Tibetan medicine’s fifteen-flavor black pill and fecal bacterial supernatant can significantly regulate the intestinal metabolism of weaned Tibetan piglets. The therapeutic efficacy of these additives in clinics may be related to the regulation of the organism’s metabolic pathways, particularly the metabolic pathway of tryptophan.

## 5. Conclusions

In summary, our findings suggest that the feeding of fecal bacterial supernatant and the fifteen-flavor black pill of Tibetan medicine improve the growth performance of weaned piglets and certain blood physiological indexes, such as white blood cells and hemoglobin. We also speculate that these improvements are due to the influence on microbial metabolites like 3-hydroxy-octylaminobenzoic acid (3-HAA) and metabolic pathways like tryptophan metabolism. These effects may lead to a reduction in diarrhea rates, enhancement of immune ability, and increased growth performance of weaned Tibetan piglets. Moreover, our results indicate that the fifteen-flavor black pill of Tibetan medicine and the anti-feeding of fecal bacterial supernatant exhibit a certain substitutive effect. This reveals a potential for the substitution products to be scientifically evaluated and selected, bringing real benefits to the relevant industry’s development.

## Figures and Tables

**Figure 1 metabolites-14-00138-f001:**
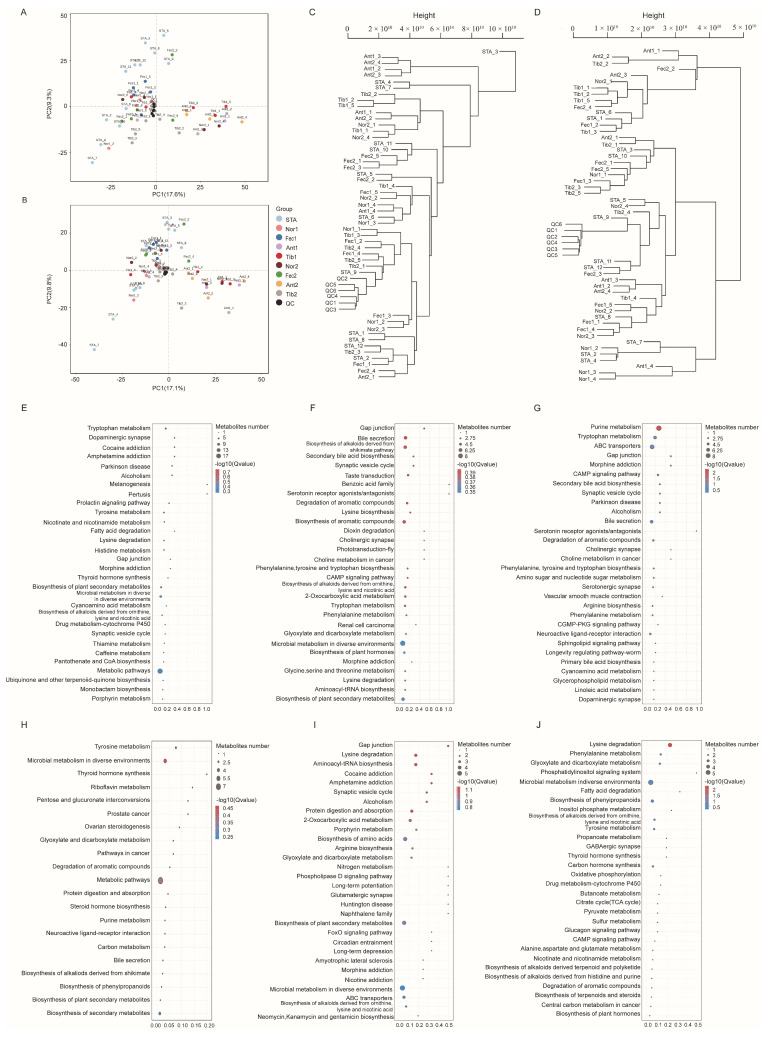
(**A**) PCA score plot for the ten groups analyzed in the positive ion mode. (**B**) PCA score plot for the ten groups analyzed in the negative ion mode. (**C**) Sample dendrogram of the 10 groups analyzed in positive ion mode. (**D**) Sample dendrogram of the 10 groups analyzed in negative ion mode. (**E**–**J**) KEGG significance bubble plot.

**Figure 2 metabolites-14-00138-f002:**
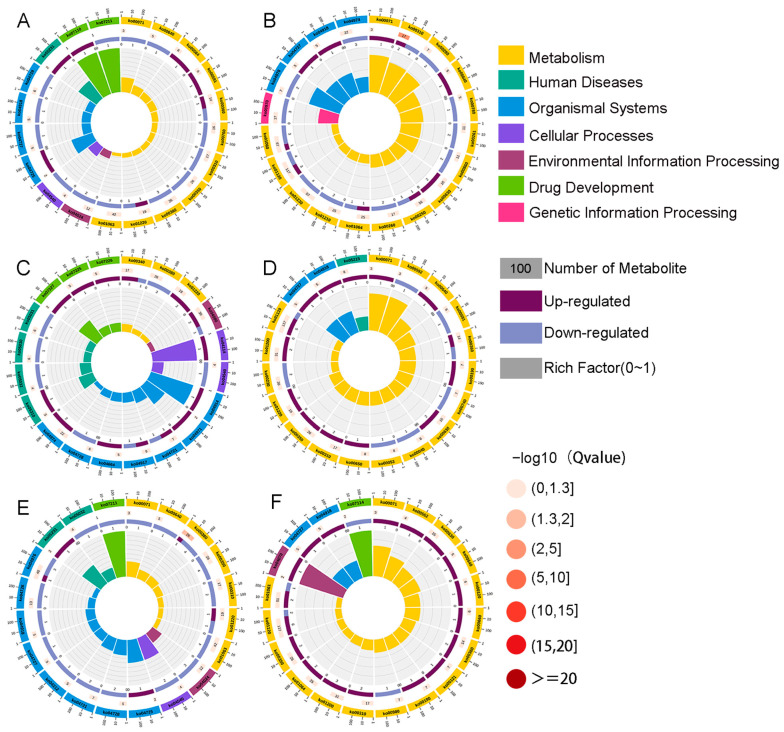
(**A**–**F**) Metabolite KEGG enrichment circle plot analysis.

**Table 1 metabolites-14-00138-t001:** Feed formulation and nutrient composition of diets.

Items	Proportion/%
Corn	45.00
Wheat bran	35.00
Soybean meal	16.00
Additive premixes 2624	4.00
DE (Mcal/kg)	2.8487
CP (%)	15.89
CF (%)	3.94
Ca (%)	0.70
TP (%)	0.66
NPP (%)	0.30
Lys (%)	0.79
Met + Cys (%)	0.53
Thr (%)	0.58
Trp (%)	0.21

Note: (1) The premix provides, per kg of concentrate supplement, the following: vitamin A 10,000 IU, vitamin D32 550 IU, vitamin E 20 IU, biotin 0.06 mg, copper sulfate 22 mg, iron sulfate 94 mg, manganese sulfate 80 mg, zinc sulfate 88 mg, potassium iodide 0.75 mg, sodium selenite 0.50 mg, calcium 0.35%, phosphorus 0.125%, and sodium chloride 0.80%. (2) The digestible energy in the table is the calculated value, and the other nutrients are measured values.

**Table 2 metabolites-14-00138-t002:** Effect of different feed additions on growth performance of weaned piglets 0–42 d after the start of the trial (*n* = 7).

Groups	Initial Weight (kg)	Diarrhea Rate in Piglets (%)	Average Daily Weight Gain (ADG, kg)	Average Daily Feed Intake (ADFI, kg)	Feed-to-Weight Ratio (F/G)	Final Weight (kg)
Nor	5.42 ± 0.06	10.5% ± 0.01 ^A^	0.037 ± 0.00 ^C^	0.414 ± 0.25	11.19 ± 0.11 ^A^	6.97 ± 0.47 ^A^
Ant	5.43 ± 0.07	7.8% ± 0.00 ^B^	0.083 ± 0.00 ^A^	0.451 ± 0.00	5.43 ± 0.36 ^B^	8.93 ± 0.09 ^B^
Tib	5.40 ± 0.03	8.2% ± 0.00 ^B^	0.089 ± 0.00 ^A^	0.449 ± 0.03	5.04 ± 0.04 ^B^	9.13 ± 0.35 ^B^
Fec	5.48 ± 0.09	8.2% ± 0.00 ^B^	0.06 ± 0.00 ^B^	0.413 ± 0.02	6.88 ± 0.18 ^C^	8.00 ± 0.10 ^C^

Note: Data in the same column labeled with no letter or with the same letter on the shoulder indicate a non-significant difference (*p* > 0.05), and a different uppercase letter indicates a highly significant difference (*p* < 0.01).

**Table 3 metabolites-14-00138-t003:** Effect of different feed additions on growth performance of weaned piglets 0–21 d after the start of the trial (*n* = 7).

Groups	Initial Weight (kg)	Diarrhea Rate in Piglets (%)	Average Daily Weight Gain (ADG, kg)	Average Daily Feed Intake (ADFI, kg)	Feed-to-Weight Ratio (F/G)	21-Day Weight (kg)
Nor	5.42 ± 0.06	9.5% ± 0.01 A	0.013 ± 0.00 C	0.37 ± 0.04	28.46 ± 0.09 A	5.70 ± 0.09 a
Ant	5.43 ± 0.07	6.8% ± 0.00 B	0.053 ± 0.00 A	0.42 ± 0.01	7.92 ± 0.17 B	6.53 ± 0.17 b
Tib	5.40 ± 0.03	6.8% ± 0.00 B	0.054 ± 0.00 A	0.40 ± 0.01	7.59 ± 0.24 B	6.53 ± 0.24 b
Fec	5.48 ± 0.09	7.0% ± 0.00 B	0.038 ± 0.00 B	0.37 ± 0.02	9.74 ± 0.10 C	6.07 ± 0.10

Note: Data in the same column labeled with no letter or with the same letter on the shoulder indicate a non-significant difference (*p* > 0.05), a different lowercase letter indicates a significant difference (*p* < 0.05), and a different uppercase letter indicates a highly significant difference (*p* < 0.01).

**Table 4 metabolites-14-00138-t004:** Effect of different feed additions on growth performance of weaned piglets 21–42 d after the start of the trial (*n* = 7).

Groups/Growth Performance	21-Day Weight (kg)	Diarrhea Rate in Piglets (%)	Average Daily Weight Gain (ADG, kg)	Average Daily Feed Intake (ADFI, kg)	Feed-to-Weight Ratio (F/G)	21-Day Weight (kg)
Nor	5.70 ± 0.09 a	10.2% ± 0.00 A	0.060 ± 0.00 A	0.460 ± 0.00	7.67 ± 0.16 A	6.97 ± 0.24 A
Ant	6.53 ± 0.17 b	7.5% ± 0.00 B	0.114 ± 0.02 B	0.475 ± 0.02	4.17 ± 0.26 C	8.93 ± 0.15 B
Tib	6.53 ± 0.24 b	6.8% ± 0.00 B	0.124 ± 0.02 B	0.450 ± 0.02	3.62 ± 0.15 C	9.13 ± 0.11 B
Fec	6.07 ± 0.10	7.5% ± 0.00 B	0.092 ± 0.01 B	0.459 ± 0.01	4.98 ± 0.06 B	8.00 ± 0.11 C

Note: Data in the same column labeled with no letter or with the same letter on the shoulder indicate a non-significant difference (*p* > 0.05), a different lowercase letter indicates a significant difference (*p* < 0.05), and a different uppercase letter indicates a highly significant difference (*p* < 0.01).

**Table 5 metabolites-14-00138-t005:** Hematopoietic index data at 21 d after the start of the trial (n = 7).

Norm	Unit	Nor1	Ant1	Tib1	Fec1
WBC	109/L	19.67 ± 0.43 ^a^	28.03 ± 0.57 ^b^	27.83 ± 0.22 ^b^	29.23 ± 0.93 ^b^
RBC	1012/L	7.96 ± 0.20 ^a^	8.66 ± 0.18 ^b^	8.12 ± 0.31 ^b^	8.65 ± 0.20 ^b^
HGB	g/L	133.00 ± 0.58 ^a^	148.67 ± 1.45 ^b^	147.33 ± 8.09 ^b^	145.33 ± 0.88 ^b^
HCT	%	42.90 ± 0.46	44.80 ± 1.16	42.90 ± 1.30	44.67 ± 0.20
MCV	fL	54.73 ± 0.73	54.77 ± 0.24	55.43 ± 0.07	53.77 ± 0.20
MCH	Pg	17.56 ± 0.30	17.57 ± 0.07	17.77 ± 0.09	17.40 ± 0.06
MCHC	g/L	322.08 ± 1.92	322.67 ± 1.76	322.00 ± 2.52	321.67 ± 1.73
RDW	%	16.23 ± 0.46	16.83 ± 1.30	16.10 ± 0.10	17.30 ± 0.50
MPV	fL	7.93 ± 0.27	7.40 ± 0.35	7.70 ± 0.06	7.47 ± 0.23
PDW	%	16.63 ± 0.15	16.63 ± 0.13	16.50 ± 0.15	16.50 ± 0.10
PCT	%	0.32 ± 0.04	0.25 ± 0.01	0.30 ± 0.01	0.30 ± 0.05

Note: Peer data labeled with no letter or with the same letter on the shoulder indicate a non-significant difference (*p* > 0.05), while different lowercase letters indicate a significant difference (*p* < 0.05).

**Table 6 metabolites-14-00138-t006:** Hematopoietic index data on day 42 after the start of the trial (*n* = 7).

Norm	Unit	Nor2	Ant2	Tib2	Fec2
WBC	109/L	26.40 ± 1.65	27.05 ± 0.50	27.10 ± 0.86	28.93 ± 2.77
RBC	1012/L	7.65 ± 0.05	8.13 ± 0.37	7.41 ± 0.31	7.61 ± 0.17
HGB	g/L	130.75 ± 2.00	134.00 ± 6.03	132.75 ± 4.67	131.00 ± 5.03
HCT	%	39.47 ± 0.97	41.17 ± 1.52	42.10 ± 1.50	41.23 ± 1.24
MCV	fL	55.27 ± 2.15	56.40 ± 1.18	56.03 ± 0.30	55.07 ± 0.98
MCH	Pg	17.83 ± 1.26	18.27 ± 0.45	18.00 ± 0.15	17.37 ± 0.60
MCHC	g/L	325.00 ± 1.00	325.00 ± 0.58	325.33 ± 0.88	326 ± 3.06
RDW	%	18.10 ± 0.45	17.83 ± 0.83	18.80 ± 0.00	17.73 ± 0.30
MPV	fL	8.07 ± 0.09	8.23 ± 0.30	8.10 ± 0.44	8.37 ± 0.12
PDW	%	16.47 ± 0.03	16.50 ± 0.23	16.43 ± 0.18	16.30 ± 0.06
PCT	%	0.33 ± 0.07	0.24 ± 0.01	0.35 ± 0.02	0.25 ± 0.04

**Table 7 metabolites-14-00138-t007:** OPLS-DA model.

Group	R2X (cum)	R2Y (cum)	Q2 (cum)
Nor1 vs. Ant1	0.669	0.994	0.955
Nor1 vs. Tib1	0.722	0.995	0.964
Nor1 vs. Fec1	0.482	0.983	0.871
Ant1 vs. Tib1	0.601	0.978	0.767
Ant1 vs. Fec1	0.597	0.998	0.955
Tib1 vs. Fec1	0.587	0.996	0.910
Nor2 vs. Ant2	0.572	0.967	0.569
Nor2 vs. Tib2	0.578	0.976	0.518
Nor2 vs. Fec2	0.533	0.981	0.549
Ant2 vs. Tib2	0.668	0.979	0.700
Ant2 vs. Fec2	0.721	0.990	0.799
Tib2 vs. Fec2	0.714	0.902	0.744

**Table 8 metabolites-14-00138-t008:** Differential metabolites of Nor1 vs. Ant1.

Metabolites	VIP	*p* Value	FC	Trend
Stearamide	2.22	0.009	7.76	up
Choline	3.11	0.001	7.18	up
1-Stearoylglycerol	1.12	0.005	6.94	up
trans-Petroselinic Acid	3.19	0.008	5.51	up
L-(-)-Malic acid	1.76	0.007	3.89	up
4-Nitrophenyl 2-acetamido-2-deoxy-α-D-glucopyranoside	5.89	0.002	2.44	up
4-Hydroxybenzoic acid	5.36	0.001	2.24	up
Dihydrozeatin	3.52	0.002	1.90	up
Panaxatriol	1.34	0.009	1.60	up
Stearic acid	1.72	0.004	1.36	up
2,8-Quinolinediol	1.89	0.008	15.31	down
trans-3-Hexenoic acid	1.40	0.009	10.96	down
Methionine	1.61	0.004	10.36	down
Lithocholic acid	1.54	0.000	8.28	down
3-Hydroxy-3-methylbutanoic acid	4.68	0.000	6.86	down
N-Acetyl-D-alloisoleucine	3.95	0.003	4.10	down
N-Acetyl-L-leucine	3.75	0.000	3.50	down
4-Hydroxybutyric acid (GHB)	2.33	0.007	3.35	down
Isoquinoline	2.30	0.008	2.90	down
Indole-3-acetic acid	2.75	0.008	2.78	down

**Table 9 metabolites-14-00138-t009:** Differential metabolites of Nor1 vs. Tib1.

Metabolites	VIP	*p* Value	FC	Trend
Bullatine A	8.14	0.000	1.39	up
Prostaglandin E2-1-glyceryl ester	6.19	0.009	6.80	up
4-Nitrophenyl 2-acetamido-2-deoxy-α-D-glucopyranoside	5.68	0.005	2.30	up
(+)12(13)-DiHOME	3.51	0.007	4.29	up
19(R)-hydroxy Prostaglandin A2	2.83	0.000	1.14	up
PC (20:3e/22:5)	2.82	0.000	1.11	up
Choline	2.68	0.006	6.13	up
3-Methylglutaric acid	2.43	0.000	1.95	up
N1-(2-piperidinophenyl)-3,4,5-trimethoxybenzamide	2.24	0.001	1.09	up
2-Oxindole	1.83	0.009	5.53	up
4-Hydroxybenzaldehyde	1.22	0.004	4.38	down
L-Phenylalanine	1.24	0.008	12.15	down
Methionine	1.31	0.002	10.13	down
Norleucine	1.82	0.003	1.04	down
DL-Tryptophan	1.86	0.007	8.47	down
Indole-3-acrylic acid	1.87	0.007	8.61	down
N1-[4-hydroxy-6-(methoxymethyl)pyrimidin-2-yl]acetamide	1.88	0.003	2.09	down
N-Isobutyrylglycine	2.01	0.002	1.46	down
5-amino-1-phenyl-1H-pyrazole-4-carbohydrazide	2.06	0.009	1.80	down
(2-oxo-2,3-dihydro-1H-indol-3-yl)acetic acid	2.11	0.000	1.39	down

**Table 10 metabolites-14-00138-t010:** Differential metabolites of Nor1 vs. Fec1.

Metabolites	VIP	*p* Value	FC	Trend
FAHFA (22:4/3:0)	2.94	0.005	4.34	up
gamma-Glutamylleucine	1.14	0.001	2.82	up
N-Acetylhistamine	1.53	0.001	2.70	up
2-Methylnicotinamide	1.55	0.001	2.70	up
(+/−)12(13)-DiHOME	1.51	0.005	2.64	up
Lactitol	4.87	0.000	1.28	up
MGDG (19:2/18:4)	1.25	0.003	1.17	up
L-Argininosuccinate	1.86	0.000	1.16	up
3-Hydroxyanthranilic acid	2.89	0.007	1.08	up
2,8-Quinolinediol	5.42	0.000	26.55	down
L-Tyrosine	2.03	0.005	14.97	down
N-Benzylformamide	2.12	0.002	11.26	down
4-Hydroxybenzaldehyde	2.52	0.000	7.87	down
5-Hydroxyindole-3-acetic acid	2.30	0.009	5.10	down
5-Hydroxyindole	4.39	0.001	4.39	down
Dopamine	3.50	0.000	2.55	down
(2-oxo-2,3-dihydro-1H-indol-3-yl)acetic acid	2.08	0.000	1.99	down
N1-(2,3-dihydro-1,4-benzodioxin-6-yl)acetamide	2.40	0.003	1.94	down
Glucuronic acid-3,6-lactone	2.20	0.003	1.85	down

**Table 11 metabolites-14-00138-t011:** Differential metabolites of Nor2 vs. Ant2.

Metabolites	VIP	*p* Value	FC	Trend
ERH	5.95	0.001	1.28	up
3,3-Dimethylglutaric acid	1.11	0.000	1.72	up
Phenethyl isothiocyanate	0.71	0.005	1.08	down
Ethylmalonic acid	1.03	0.009	2.23	down
D-(−)-Lyxose	1.19	0.001	1.70	down
N6-Acetyl-L-lysine	1.78	0.000	5.17	down
(+/−)12(13)-DiHOME	2.00	0.008	13.11	down
11-Epiprostaglandin E1	2.04	0.001	5.08	down
Dehydrodiisoeugenol	2.89	0.004	1.03	down

**Table 12 metabolites-14-00138-t012:** Differential metabolites of Nor2 vs. Tib2.

Metabolites	VIP	*p* Value	FC	Trend
Bullatine A	8.00	0.000	2.72	up
Piperine	5.44	0.001	1.07	up
D-Ribulose 1,5-bisphosphate	2.91	0.006	2.23	up
3-(3-bromo-4-hydroxy-5-methoxyphenyl)-2-cyanoacrylamide	2.78	0.007	3.70	up
Cyanidin	1.51	0.005	1.08	up

**Table 13 metabolites-14-00138-t013:** Differential metabolites of Nor2 vs. Fec2.

Metabolites	VIP	*p* Value	FC	Trend
N-(3-chloro-2-methylphenyl)-N′-(3-methoxypropyl)thiourea	6.17	0.000	1.17	up
Lactitol	5.32	0.002	1.88	up
3-Iodo-L-tyrosine	4.47	0.042	2.14	up
L-Threonic acid-1,4-lactone	3.36	0.048	3.79	up
D-Ribulose 1,5-bisphosphate	3.28	0.005	1.90	up
(+/−)-CP 47,497-C7-Hydroxy metabolite	3.23	0.044	2.77	up
3-(3-bromo-4-hydroxy-5-methoxyphenyl)-2-cyanoacrylamide	2.74	0.030	2.51	up
Cyanidin	2.08	0.001	1.09	up
Acetophenone	1.15	0.034	23.03	up
FAHFA (8:0/18:3)	1.14	0.043	1.41	up
5(S),15(S)-DiHETE	1.23	0.039	1.59	down
3-(propan-2-yl)-octahydropyrrolo[1,2-a]pyrazine-1,4-dione	1.41	0.004	1.18	down
Ribitol	1.57	0.000	1.05	down
Uric acid	3.70	0.047	5.29	down

## Data Availability

The data presented in this study are available on request from the corresponding author.

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
