# Peer review of "Effects of Different Feed Additives on Intestinal Metabolite Composition of Weaned Piglets"

_metabolites, 2024, doi:10.3390/metabo14030138_

Round 1
Reviewer 1 Report
Comments and Suggestions for Authors
Dear Authors,
Thank you for preparing this mansukript. It ist written in good and understandable English and easy to follow. Please find my comments per line and some questions below:
Line 17: How many animals? How long did you measure?
Line 19/20: please define them as group names
Line 26/57/107: Fifteen Black Herbal Pills of Tibetan Medicine: please give the exact composition and the producer. Is this permitted for use in animals?
Line47: please give literature for this
Line 110: please give a space between “ginger.In”
Line 123/126: “the beginning and end of each phase” as you stated would be day 0, 21, (22), 42; The end day of one phase is also the start day of the next phase? Please explain and clarify!
Line 141: Please ad some information about the fasting: How long? Only food or water too? Bedding material removed?
Materials and methods: please explain how often did you check for diarrhea? How often did you feed the piglets? Did you check what your faecal bacterial supernatant consists of? Please give the composition!
Table 3: Why didn´t you use two/or three- way ANOVA with the post test accordingly to calculate differences?
Figure 2: I am sorry, but I cannot read what is written outside the coloured circles. At magnification the letters are blurred and I cannot find the Qvalve. Maybe you can offer this at higher resolution for publication.
Results: A Microbiome analysis would be great to see which microbiota shift where. This should be included into analysis. Maybe this would allow matching metabolites and microbiota.
Please give the animal experiment approval number!
kind regards!
Author Response
Thank you for taking the time to review our article, and we greatly appreciate your valuable feedback.
1.Line 17: We selected 28 healthy 30-day-old Tibetan weaned piglets with basically the same weight and randomly divided them into 4 treatments, with 7 piglets in each replicate.The pigs were fed these diets for a duration of 42d, which were divided into two phases: phase 1 (0d to 21d) and phase 2 (21d to 42d).
2.Line 19/20::The control group (basal feed without any antibiotic additions) (Nor), the group with the addition of antibiotic lincomycin (Ant), the group with the addition of fifteen-flavour black pills of Tibetan medicine (Tib), and the group with the addition of fecal bacterial supernatant (Fec).
3.Line 26/57/107: Tibetan medicine 15-flavor black pills, Tibetan medicine name: Tamen Jiuaribu prescription was made in the 14th century BC, which belongs to Chinese patent medicine preparation.The medicine is prepared from mirabilite, salt (parched), Rhododendron fortunei, nutmeg, Akebia stem, coriander fruit, mirabilite, sal ammoniac, and light.Fifteen kinds of pure natural mineral and plant medicines such as salt, purple sal, bangga, Tibetan costustoot, long pepper, black pepper and dried gingerResearch in recent years
It is found that the Tibetan medicine Shiweihei Pill has the functions of protecting gastric mucosa, improving the movement tension of gastric muscle strips, promoting gastric emptying and accelerating.In addition to this mechanism, studies have shown that Tibetan medicine Shiweihei Pill can also effectively increase the stomach of gastritis patients.Mucosal blood flow, improve microcirculation, soften proliferative lesions and promote their absorption. This medicine is combined with other Tibetan medicines to treat atrophy.Sexitis has achieved good curative effect.
Your input is invaluable in helping us improve the quality of our work, and we are committed to making the necessary revisions as per your suggestions. Once again, we would like to extend our gratitude for your constructive feedback, and we look forward to resubmitting the revised article for your further evaluation.
4.Line 47: The reference references has been inserted.
5.Line 110: Thank you for your reminder, I have added spaces in the article.
6.Line 123/126: The pigs were fed these diets for a duration of 42d, which were divided into two phases: phase 1 (0d to 21d) and phase 2 (21d to 42d), The end day of phase 1 is also the start day of the phase 2.
7.Line 141: Fasting for 8 hours before blood collection can not stop water and keep the original feeding environment.
8.Materials and methods: Check the diarrhea of weaned piglets every day and ensure that piglets can eat freely every day.
9.Table 3: Thank you for your suggestion. Can I keep the original calculation method.
10.Figure 2: Thank you for your review, I have improved the resolution of the picture.
11.Results: Thank you for your suggestion. Some of the differences have been moved to the discussion.
12. All experiments in this manuscript were approved by the Animal Welfare and Reseach Ethics Committee of Tibet Agriculture and Animal Husbandry University (approval number: XK624), approved by 5 January 2023.

Reviewer 2 Report
Comments and Suggestions for Authors
Potentially very interesting Manuscript, but since it has certain shortcomings in the presentation of research to the wider scientific public, the Manuscript needs to be adapted for publication.
All comments and remarks to the Authors are written within the Manuscript in the enclosed pdf file.

Author Response
Thank you for taking the time to review our article, and we greatly appreciate your valuable feedback.Your input is invaluable in helping us improve the quality of our work, and we are committed to making the necessary revisions as per your suggestions. Once again, we would like to extend our gratitude for your constructive feedback, and we look forward to resubmitting the revised article for your further evaluation.

Reviewer 3 Report
Comments and Suggestions for Authors
Dear authors,
it is positive that this manuscript “Effects of Lincomycin and Faecal Bacterial Stress on Intestinal Metabolite Composition of Weaned Tibetan Piglets” (metabolites-2807796) by Mingxuan Zhao and coauthors devoted to the evaluation of the effects of feeding on weaning stress in Tibetan piglets. The authors used the following four “dietary treatments”: Lincomycin hydrochloride (group Ant), Chinese medicine preparation (group Tib) and faecal bacterial supernatant (group Fec) in comparison to the control (group Nor). It is especially interesting the usage of a composition of the fifteen pure natural mineral and botanical medicines, i.e. so-called the “Fifteen Black Herbal Pills of Tibetan Medicine” (FBHPTM). The authors found that the addition of the faecal bacterial supernatant and the FBHPTM (in the feeding diets) improved the levels of particular metabolites, some blood parameters (such as white blood cell counts and haemoglobin level) and the growth performance of weaned Tibetan piglets. it is important that the incidence of diarrhea was significantly lower (P<0.01), but the average daily gain (ADG) was significantly higher (P<0.01), in the Ant, Tib and Fec groups compared to the Nor group. A pronounced influence of these “dietary treatments” on some metabolic pathways of weaned piglets (like metabolism of tryptophan, some purines and nucleotides) was shown.
The manuscript analyze the literature works in detail and at high level of discussion. I do not doubt the technical quality of the work and feel that there is a sufficient impact on a broader readership to justify publication in the "Metabolites". This topic is in frame of the journal scope, the subject matter is treated in depth. Thus, the present manuscript is actual and important, especially in the field of the development of efficient antibiotic replacement products.
There are some comments:
1. In the part 2.1. “Animals, Design and Diets” in 2. “Materials and Methods” (page 3, lines 101-113) the authors wrote “Twenty-eight weaned Tibetan pigs …. The trial consisted of four treatment groups: the control group (Nor1/2 group), the 105 lincomycin group (Ant1/2 group), the faecal bacterial supernatant group (Fec1/2 group), and the Tibetan medicine fifteen-flavour black pill group (Tib1/2 group)”. Probably, the amount of the animals in each group was 7 (n=7), but in each 1/2group – 3 or 4 animals (n=3 or 4) !? There were no definitions concerning the meaning of the “1/2 group”, as well as the amount of the animals in each group and “1/2 group” that had to be included in the part 2.1. “Animals, Design and Diets” (2. “Materials and Methods”).
2. In the part 3.1. “Growth Performance“ in 3. “Results” (page 6, lines 199-200) a format have to be corrected in the “title-line” in the all columns in the Table 1. “Effect of different feed additions on growth performance of weaned piglets 0-42d after the start of the trial (n=7)”.
3. In the part 3.2. “Comparison of Blood Physiological Indicators” (page 6, lines 201-212) there are names of the groups are the following: Ant1, Tib1, Fec1 and Nor1 groups (on 21d of the experiment), as well as in the Table 2. “Haematopoietic index data on 21d after the start of the trial (n=7)”. It is important to include the additional Table with the blood physiological indices in weaned piglets on 42d of the experiment (probably groups Ant2, Tib2, Fec2 and Nor2), even they do “not reach statistical significance”. So, it can be two tables: Table 2a and Table 2b, because all obtained data are important. The authors can include the additional Table 2b (with the blood physiological indices in weaned piglets on 42d of the experiment) in the part “Supplementary materials” as another option.
4. In the part 3.2 “Comparison of Blood Physiological Indicators” (page 6, lines 206-207) the authors wrote “….the leukocyte count in the Tib1 and Fec1 groups was significantly higher than that in the Nor1 group (P<0.01)”. But there is no mention on the Ant1 group in spite of the same high leukocyte count as in the Tib1 and Fec1 groups.
In the part 4. “Discussion” (page 17, lines 328-329) the authors wrote “Changes in the number of leukocytes can indicate the level of cellular immunity [28-29]. The Tib group of weaned piglets exhibited significantly higher leukocyte counts than the Nor group on the 21st day of the test (P < 0.01)”. But there is no real discussion given of this negative effect for Tib group (or may be also for Tib1 and Tib2 groups ?). There is no mention on the Fec1 and Ant1 groups in spite of the same high leukocyte counts. The complete discussion of this negative effect for Tib, Fec and Ant groups must be given in the part 4. “Discussion” at page 17.
Accept after minor revision (corrections to minor methodological errors and text editing).
Minor editing of English language required.
Comments on the Quality of English LanguageAuthor Response
Thank you for taking the time to review our article, and we greatly appreciate your valuable feedback.
- In the part 2.1. Thank you for finding the problem. Group 1 represents the first stage of feeding (0 d-21 d), and group 2 represents the second stage of feeding (21 d-42 d).
- In the part 3.1. Thank you for your careful approval,I have completed the modification in the text
- In the part 3.2.Thank you for your careful approval. I have joined it in the attachment.
- In the part 3.2Thank you for your careful approval,I have revised it in the text
In the part 4.White blood cells are one of the important cells in the body, can play the role of regulating immunity, can resist pathogens and bacteria, reduce the probability of the body infection. The Tib group has leukocytosis in the normal range, so it can be inferred that taking Tibetan medicine 15 black pills can improve the immunity of weaned piglets, thus reducing the occurrence probability of diarrhea and other diseases
Your input is invaluable in helping us improve the quality of our work, and we are committed to making the necessary revisions as per your suggestions. Once again, we would like to extend our gratitude for your constructive feedback, and we look forward to resubmitting the revised article for your further evaluation.

Round 2
Reviewer 1 Report
Comments and Suggestions for Authors
DearAuthors,
Thank you for taking my comments into Account.
The only questions left open ist the exact composition of the fifteen flavor black pills in % for example.
Kind regards
Author Response
Thank you for taking the time to review our article, and we greatly appreciate your valuable feedback. The Tibetan medicine 15 black pill belongs to Chinese patent medicine preparation, composed of 75g of cold stone, 75g of salt, 75g rhododendron, 15g nutmeg, g 75g of clematis, 50g coriander fruit, 20g fire nitrate, 20g, 20g bright salt, 50g, 50g, 25g piper, 25g black pepper, 40g dried ginger, with honey 50g, after drying.Your input is invaluable in helping us improve the quality of our work, and we are committed to making the necessary revisions as per your suggestions. Once again, we would like to extend our gratitude for your constructive feedback, and we look forward to resubmitting the revised article for your further evaluation.

Reviewer 2 Report
Comments and Suggestions for Authors
Unfortunately, no changes were made to the Manuscript as suggested. The main complaint is still that there is no order in the analyzed parameters. In other words, the order should be the same through Methods, Results, and Discussion. Again, some parameters just appear without being mentioned before. For example, the parameter mentioned in the Conclusion (hemoglobin) is nowhere mentioned in the Methods. Also in Methods, chapter 2.6. is still without any reference. Through the Results (3.4.) and the Discussion (pdmah in the first sentence) Multivariate statistical analysis is discussed, but it is not mentioned in the Methods. In general, there is still work to be done on the structure of the Manuscript. The first sentence of the Abstract remained the same and does not follow the changes made by the Authors in the Manuscript.
Apart from the aforementioned, the title is unclear, "... and faecal bacterial stress on ..." you probably meant the influence of supplements on weaning stress in piglets, please write it like that.
In short, I cannot yet recommend the Manuscript for publication.
Author Response
Thank you for taking the time to review our article, and we greatly appreciate your valuable feedback.
- The main complaint is still that there is no order in the analyzed parameters. In other words, the order should be the same through Methods, Results, and Discussion.Again, some parameters just appear without being mentioned before. For example, the parameter mentioned in the Conclusion (hemoglobin) is nowhere mentioned in the Methods. Also in Methods, chapter 2.6. is still without any reference. Through the Results (3.4.) and the Discussion (pdmah in the first sentence) Multivariate statistical analysis is discussed, but it is not mentioned in the Methods.
- In general, there is still work to be done on the structure of the Manuscript. The first sentence of the Abstract remained the same and does not follow the changes made by the Authors in the Manuscript.
- Apart from the aforementioned, the title is unclear, "... and faecal bacterial stress on ..." you probably meant the influence of supplements on weaning stress in piglets, please write it like that.
Question reply:
- Thanks for your modification suggestions, I have added the parameter of hemoglobin in the method, and I have included a description of multivariate statistical analysis inMethods, chapter 2.6.
- Thanks for your modification suggestions, I amend the first sentence of the abstract as"To study the effects of different feed additives on weaning stress of Tibetan piglets ".
- Thanks for your careful review, I have modified the title to"Effects of different feed additives on intestinal metabolite composition of weaned piglets ".
Your input is invaluable in helping us improve the quality of our work, and we are committed to making the necessary revisions as per your suggestions. Once again, we would like to extend our gratitude for your constructive feedback, and we look forward to resubmitting the revised article for your further evaluation.

Round 3
Reviewer 2 Report
Comments and Suggestions for Authors
The authors have accepted the proposed changes and corrections, and now they can approve the manuscript for the further procedure of the Editorial Board.